# Study on the Pyrolysis Kinetics and Mechanisms of the Tread Compounds of Silica-Filled Discarded Car Tires

**DOI:** 10.3390/polym12040810

**Published:** 2020-04-04

**Authors:** Chuansheng Wang, Baishun Zhao, Xiaolong Tian, Kongshuo Wang, Zhongke Tian, Wenwen Han, Huiguang Bian

**Affiliations:** Key Laboratory of Advanced Manufacturing Technology of Polymer Materials in Shandong, College of Electromechanical Engineering, Qingdao University of Science and technology, Qingdao 266061, Shandong, China

**Keywords:** pyrolysis, kinetics, pyrolysis mechanism, thermogravimetry, multi-kinetic method

## Abstract

The disposal of used automobile tires is a major waste concern. Simply stacking tires and allowing them to decompose will harbor breeding mosquitoes that spread viruses, whereas burning them will release acidic and toxic gases. Therefore, one viable option is pyrolysis, where elevated temperatures are used to facilitate the decomposition of a material. However, the lack of theoretical support for pyrolysis technology limits the development of the pyrolysis industry when it comes to discarded tires. The purpose of this research is to put forward a brand-new multi-kinetic research method for studying materials with complex components through the discussion of various kinetic research methods. The characteristic of this kinetic research method is that it is a relatively complete theoretical system and can accurately calculate the three kinetic factors considered during the pyrolysis of multicomponent materials. The results show that the multi-kinetic research method can obtain the kinetic equation and reaction mechanism for the pyrolysis of tires with high accuracy. The pyrolysis process of this compound was divided into two stages, Reaction I and II, where the kinetic equation of Reaction I was $f\left(\alpha \right)=0.2473\alpha^{-3.0473}$, with an activation energy of 155.26 kJ/mol and a pre-exponential factor of 5.88 × 10^9^/min. Meanwhile, the kinetic equation of Reaction II was $f\left(\alpha \right)=0.4142\left(1-\alpha \right){\left[-\ln\left(1-\alpha \right)\right]}^{-1.4143}$, while its activation energy was 315.40 kJ/mol and its pre-exponential factor was 7.86 × 10^17^/min. Furthermore, based on the results of the research analysis, the reaction principles corresponding to Reaction I and Reaction II in the pyrolysis process of this compound were established.

## 1. Introduction

Rubber can be used in many aspects of industry and beyond thanks to its excellent properties such elasticity, reversible deformation, insulation, and wear resistance after being processed. According to the International Rubber Group (IRSG), almost 60% of global consumption (about 30.12 million tons in 2019) is attributed to the world’s tire manufacturing industry, with nearly 18 million tons of waste tires produced each year [1].

However, due to factors such as oxidation, wear, and mechanical damage, a significant number of tires are becoming waste items. Increasing ‘black pollution’ from disposed tires has caused tremendous pressure and damage to the world’s ecological environment. Therefore, the problem of finding a solution to treat the waste tires has attracted the attention of research groups around the world [2,3,4,5]. In the past, the primary ways of reusing or disposing waste tires were either retreading, incinerating, or disposing in a landfill. Nowadays, pyrolysis is considered to have the most potential for an environmentally friendly disposal. Due to the extremely difficult degradation of waste tires, disposing them in landfills will occupy a large amount of land resources and even lead to the breeding of bacteria. Incineration of waste tires is also not a viable answer, as doing so would release acidic gas and a large amount of small-particle-sized dust into the air, thus seriously polluting the environment. One advantage is that the pyrolysis of waste tires can overcome the shortcomings of landfilling and incineration. In addition, this method can also be used to recover high value-added recycled materials with high economic benefits [6,7]. Lastly, pyrolysis has good comprehensive environmental benefits and can save resources. 

Thermogravimetric analysis and multi-kinetics research methods have been of recent focus for the treatment of waste plastics and tires [8]. These methods can be used for other polymer complexes, as many other products manufactured from polymers do not degrade in nature in a short period of time and will bring great challenges to the environment. For example, the products of the pyrolysis of other types of waste plastics have been studied, such as plastic casings of television sets [9,10,11,12]. In most cases, pyrolysis is a new and useful waste treatment method because the products can often be recycled in other applications. For example, pyrolysis chars from coals can even be used as highly insulating building material [13]. Meanwhile, bio-bitumen can be obtained from organic fractions of municipal solid waste [14], while liquid hydrocarbon biofuels can be obtained from microalgae or waste cooking oil by catalytic pyrolysis [15,16], and the types and principles of catalysts are also of great significance for research [17,18]. Considering the value of the products of catalytic pyrolysis, carbon nanofibers obtained from the catalytic pyrolysis of acetylene have good development prospects [19,20]. It is even possible to recover graphite or electrode materials from spent lithium-ion batteries via pyrolysis [21,22]. In addition, natural macromolecule materials such as coconut copra and rice husk can also produce useful pyrolysis products such as biochar [23,24]. Not only is studying the products important, it is also important to consider the effects of various pyrolysis system parameters on the biomass pyrolysis process [25,26].

Previous studies have focused on the pyrolysis equipment and methods for waste compounds, or have concentrated on the pyrolysis process and products of other macromolecular substances [27,28,29]. The pyrolysis kinetics and the corresponding mechanistic model for waste tires have an important guiding significance in the overall pyrolytic process, but they are yet to be fully revealed. Therefore, it is necessary to study the process used for tires so that the parameters of the general pyrolysis process can be improved and the pyrolysis product structure can be optimized. To investigate this, the three kinetic factors (activation energy *E*, pre-exponential factor *A*, and kinetic equation *f(α*)) that describe the pyrolysis process need to be obtained.

It should be noted that the production process of the tire is complicated but precise, and it usually requires several types of composite rubber. In particular, the composition of the tread comprises a significant proportion in the whole tire, which contains two or three kinds of rubber, as well as over 10 types of additives. Additionally, the total pyrolysis process of the tread rubber from waste tires cannot be characterized clearly and accurately by merely one method [7,30,31]. Using the Kissinger–Akahira–Sunose (KAS) method to calculate the pre-exponential factor is not feasible when the reaction model/mechanism has not been determined [32,33,34]. When some optimization algorithms such as the genetic algorithm (GA) and shuffled complex evolution (SCE) are used to calculate the kinetic parameters including the pre-exponential factor, the reaction model/mechanism still needs to be assumed in advance [8]. Based on the peak differentiating analysis using a Gaussian function, the pyrolysis of waste rubber is divided into several sub-reactions. The pyrolysis kinetics can be obtained by analyzing the sub-reactions, but there is no reliable evidence for the establishment of an obtained mechanism [35]. Gonzalez et al. focused their research on product distribution, but did not determine a kinetic model and did not establish a corresponding physical model [36]. Complex reactions will overlap when the entire tire is selected as the experimental material, which is not conducive to the calculation of kinetic results. Unlike previous literature, this study first separated the compounds according to the different structures of the tires to obtain more accurate kinetic results. It should be emphasized that if the kinetic method requires the assumption of a kinetic model in advance, errors will be introduced [37,38]. Leung et al. used a single-rate scanning method to conclude that it is impossible to establish an accurate kinetic model for waste tire pyrolysis [38], and Conesa et al. also reached a similar conclusion that a fractional model is not sufficient to explain the pyrolysis of waste tires, but a model was not given to explain the first reaction [39]. At the beginning of the century, the International Thermal Analysis and Calorimetry Society (ICTAC) Kinetic Branch and multinational thermal analysts showed that using a single scan rate method to process thermal analysis kinetic data gives results that are not reliable and cannot reflect the complex nature of a solid-state reaction [40,41]. As a result, the international thermal analysis community has called for the use of multiple scan rate methods to determine thermal analysis data. In addition, as a way to determine the complex nature of the reaction, it is necessary to determine the change in activation energy with conversion using the iso-conversion method [40,41]. The multi-kinetic method integrated in this study can avoid pre-supposition models (assuming a kinetic model in advance) when studying the pyrolysis of a mixture such as waste tire tread rubber.

In summary, a more systematic and rigorous kinetics method must be adopted in dealing with the pyrolysis kinetics and the mechanistic model of a heterogeneous solid-phase material such as a tire. Therefore, a new research method must be defined to determine thermal analysis data and to reveal the complex nature of the reaction by determining the change in activation energy with conversion.

## 2. Materials and Methods

### 2.1. Materials

The waste material used in this kinetic analysis was the tread rubber and inner liner of a discarded silica-filled car tire. The reason for choosing this kind of waste is that the silica-filled tires are extensively used worldwide. It is also known that a large amount of silica is used as the tire’s reinforcing agent instead of carbon black. According to the data provided by the tire manufacturer (Tire Technology Alliance, Qingdao, China), the materials and composition ratios used in the tread rubber and inner liner formulation are listed in Table 1. Several additives are also used, such as fillers, reinforcing agents, antioxidants, and processing aids. More systematic studies will be conducted in future work regarding the effect of the mass ratios of natural rubber to butadiene rubber on the pyrolysis of the waste tire mixtures.

### 2.2. Methods

The thermogravimetric test was carried out using a thermogravimetric analyzer (TG 209 F3 Tarsus, NETZSCH, Shanghai, China; temperature range: 0 °C to 1100 °C; weight range: 0 to 2000 mg). For a given thermogravimetric instrument, the effects of balance sensitivity, sample holder, and thermocouple are fixed. Therefore, it is necessary to reduce or eliminate these system errors through mass calibration and temperature calibration before the experiment. The conclusion obtained in the experiment of thermogravimetric analysis is highly consistent with the actual pyrolysis process of the tire [42,43]. The temperature ranged from room temperature to 800 K under heating rates of 10, 15, 20, and 25 K/min, respectively. The thermogravimetric test was carried out under a nitrogen atmosphere at a gas flow rate of 50 mL/min. The heating rate and temperature range used in the testing process were consistent with the conditions used for slow solid pyrolysis during actual reaction operations [44,45,46]. The mass of the sample used for the thermogravimetric test was about 15 mg. In addition, a sufficient amount of powder was stored in a drying oven at 80 °C for 6 h to remove moisture. It has been shown in the literature that smaller sample sizes have little effect on the pyrolysis results [36,38].

### 2.3. Pyrolysis Kinetics

#### 2.3.1. Proposed Reaction Mechanism 

Pyrolysis of the tread rubber in the silica-filled tire follows the non-isothermal heterogeneous reaction kinetic equation. In general, Equation (1) is used to describe the thermal decomposition kinetics of solids.
(1)$$\frac{d\alpha }{dT}=\frac{A}{\beta }\exp\left(-\frac{E}{RT}\right)f\left(\alpha \right),$$
(2)$$\alpha =\frac{m_{0}-m}{m_{0}-m_{\infty }},$$
where α is the pyrolysis conversion rate, *T* is the pyrolysis temperature (K), β is the heating rate (K/min), and *A*, *E*, and *R* are the pre-exponential factor (S^−1^), activation energy (J/mol), and general gas constant (J/(mol·K)), respectively. f(α) is a reaction mechanism function that controls the reaction process, and m_0_, m, and m_∞_ are the respective initial, transient, and final masses (mg) of the sample during pyrolysis.

The Friedman equation, Equation (3), can be obtained by rearranging the above equations [47].
(3)$$\ln\left(\beta \frac{d\alpha }{dT}\right)=\ln\left(Af\left(\alpha \right)\right)-\frac{E}{RT}.$$

The characteristic of this method for the thermogravimetric analysis curves of multiple heating rates is as follows: If the same conversion rate α is taken, ln(Af(α)) should be a constant, and the slope of the line can be obtained by plotting ln(βdα/dT) versus 1/*T*. Thus, a relatively reliable activation energy value is obtained. However, the activation energy obtained by this method is not as accurate as that obtained by the Starink method, although if we only use the calculated activation energy to describe the trend with conversion *α* at different heating rates, it is a justified basis for judging whether the pyrolysis process conforms to a single kinetic model. If the activation energy *E* hardly changes with the conversion rate *α*, it indicates that the entire pyrolysis process follows a single kinetic model. Conversely, if the activation energy *E* changes with the conversion rate *α*, and the change shows a certain regularity, it can be used as the judgment basis for whether the kinetic model changes [47].

#### 2.3.2. Method for Solving Kinetic Parameters

(1) Solving for activation energy *E*


Although the activation energy *E* can be solved by the Friedman equation (Equation (3)), the Starink equation in the equal conversion method is considered to be more accurate than the Flynn–Wall–Ozawa (FWO) and Kissinger–Akahira–Sunose (KAS) methods [48]. Therefore, the Starink method (Equation (4)) is used to find the activation energy *E*.
(4)$$\ln\left(\frac{\beta }{T^{1.8}}\right)=-\frac{BE}{RT}+constant,$$
where *B* = 1.0037 and *R* is the universal gas constant (*R* = 8.314 J/(mol·K)). 

The different heating rates β and the temperature *T* at the same conversion rate α are substituted into the above equation. Then, the activation energy of each stage is characterized by the slope of the straight line in the $\ln\left(\beta /T^{1.8}\right)$-*B*/*RT* diagram. The advantage of this method is that the calculation error of the kinetic parameters due to the assumption f(α) can be excluded.

(2) Solving for the kinetics mechanism function f(α)

Málek et al. proposed a relatively complete thermal analysis kinetics method, where the equivalent conversion rate method is first used to obtain the activation energy *E*, and the kinetic mechanism function f(α) form is then determined from the shape. The eigenvalues of the definition function y(α) (Equation (5)) are transformed from the experimental data [49,50].
(5)$$y\left(\alpha \right)={\left(\frac{T}{T_{0.5}}\right)}^{2}\frac{\left(\frac{d\alpha }{dt}\right)}{{\left(\frac{d\alpha }{dt}\right)}_{0.5}}=\frac{f\left(\alpha \right)\times G\left(\alpha \right)}{f\left(0.5\right)\times G\left(0.5\right)},$$
where y(α) is the definition function. The commonly used kinetic mechanism functions f(α) and G(α) are shown in Table 2.

If the experimental curve overlaps with the standard curve and the linear correlation coefficient is high, indicating that the experimental data points all fall on a certain standard y(α) curve, it is determined that the corresponding f(α) or G(α) of the standard curve is the most generalized kinetics mechanism function. It should be noted that if several kinetic equations meet the above requirements, the screening method mentioned in Section 2.3.3 is used for further screening.

(3) Solving for the pre-exponential factor A

After obtaining the activation energy and the pyrolysis kinetic equation, the pre-exponential factor can be obtained using the FWO equation (Equation (6)), considering that the solution of the pre-exponential factor is deeply affected by the activation energy and the kinetic equation. If an equation that does not involve a kinetics model is used to solve the pre-exponential factor, the result will inevitably introduce an unknown factor. The FWO method, as a commonly used kinetic research method, contains the activation energy and the integral form of the kinetic model, so the pre-exponential factor obtained by this method is more accurate.
(6)$$\ln\beta =\ln\left(\frac{AE}{RG\left(\alpha \right)}\right)-5.3308-1.0516\frac{E}{RT}.$$

The kinetics mechanism function G(α) refers to Table 2.

#### 2.3.3. Test method for kinetic parameters

By integrating Equation (1) mentioned in Section 2.3.1, we obtain the following expression:(7)$$G\left(\alpha \right)=\int_{0}^{\alpha }\frac{d\alpha }{f\left(\alpha \right)}=\frac{A}{\beta }\int_{0}^{T_{\alpha }}\exp\left(-\frac{E}{RT}\right)dT=\frac{AE}{\beta R}\int_{\mu_{\alpha }}^{\infty }\frac{\exp\left(-\mu \right)}{\mu^{2}}d\mu .$$

If both sides of Equation (7) are in the logarithmic form, then it is written as
(8)$$\ln\left(G\left(\alpha \right)\right)=\left(\ln\left(\frac{AE}{R}\right)+\ln\left(p\left(\mu \right)\right)\right)-\ln\beta .$$

Considering Equation (8), if the value of α at the same temperature at multiple heating rates is put into the equation, the value of ln(AE/R)+ln(p(y)) will be a constant [51,52,53]. Using the optimization model to calculate the function G(α) may inadvertently ignore the influence of E and A on G(α), resulting in an introduced error. However, if the studied pyrolysis reaction meets a specific reaction model G(α), the slope of the curve of ln(g(α)) versus lnβ should be equal to –1 at the same temperature at multiple heating rates. The corresponding linear correlation coefficient *R*^2^ should also be equal to 1. Therefore, it is feasible to use this method to test the accuracy of the kinetic model obtained in Section 2.3.2. Additionally, as the solution method and the verification method are completely different systems, the two methods complement each other, so research methods like this are trusted and independently verified.

## 3. Results and Analysis

### 3.1. Thermogravimetric Analysis

As shown in Figure 1, 17 fitting curves of ln(βdα/dT) to 1/*T* at iso-conversion rates are shown, with conversion rates ranging from α = 0.1 to 0.9. 

The curve in Figure 1 depicts the slope of a line obtained by ln(βdα/dT) versus 1/*T*, from which the different intervals (α = 0.1–0.25 and α = 0.25–0.9) can be obtained. When α = 0.1–0.25, plotting ln(βdα/dT) vs. 1/*T* shows a relatively stable change in the slope of the line. As such, it can be concluded that in this interval, the pyrolysis process follows a certain kinetic equation, and the segment is defined as Reaction I. Similarly, at α = 0.25–0.9, the relationship between ln(βdα/dT) and 1/*T* is linear, indicating that in this α interval, the pyrolysis process also follows a certain kinetics equation, and this segment is defined as Reaction II. The gradients of the lines in the plots differ for the two different α intervals. Hence, with the sample mass and temperature corresponding to α = 0.25 as the reaction cut-off points, the conversion rate curves for Reactions I and II are plotted as shown in Figure 2c,e, respectively. The reaction rate (*dα/dT*) curves are plotted using the differential definition according the conversion rate curves, as shown in Figure 2b,d,f. For comparison, the same analysis method is used to analyze the inner liner and divide the pyrolysis process into Reaction A and Reaction B.

The situation above means that the process has undergone a change in the type of reaction, and that the entire reaction process cannot be fully described by using a single fixed kinetic equation. In fact, after computational studies, the results show that there is no single kinetic model listed in Table 2, which can describe the entire pyrolysis process, giving support to the conclusions mentioned in the introduction, where it was stated that the complete pyrolysis process of the tread rubber of waste tires cannot be characterized clearly and accurately by a single kind of method [30,31,54].

Figure 2 shows the relationship between the conversion rate α and the reaction rate dα/dT of the test sample, under a nitrogen atmosphere with heating rates of 10, 15, 20, and 25 K/min. As shown in Figure 2a, the conversion curve exhibits a hysteresis of theoretical temperature hysteresis, which is manifested as a curve shifting to a higher temperature region with the increase in heating rate. Generally, rubber products are used as insulators, and consequently, as the applied heating rate is increased, the thermal conductivity inside the test sample cannot follow the growth rate of the program temperature. In addition, the size and surface area of the test sample have an effect on the efficiency of pyrolysis. Furthermore, the test sample used in this study was only about 15 mg and was not pulverized, which differs from the industrial pyrolysis process, where entire tires are usually used for the process primarily for the purpose of avoiding the increase in process costs caused by crushing used tires. It is foreseeable that the temperature hysteresis will be particularly pronounced when the complete waste tire is subjected to pyrolysis.

According to Figure 2c,e, the same conclusion can be obtained by the conversion curves of Reaction I and Reaction II and the reaction rate curve. Following the segmentation of the total reaction, temperature hysteresis still exists in different reaction stages. In addition, analyzing the overall shape of the Reaction I conversion curve and the reaction rate curve in Figure 2c,d, it was found that the pyrolysis of the tread rubber in the silica-filled tire was likely to conform to the acceleration model in the pyrolysis type, and this can be illustrated by the power-law model shown in Table 2.

### 3.2. Kinetics Analysis

In this section, the activation energy *E* at different conversion rates α of the entire pyrolysis process was calculated using the Starink method proposed in Section 2.3.2. Figure 3 shows the change in *E* with α based on the Starink method (Figure 2a) through the pyrolysis process of the tread rubber in the silica-filled tire. 

As shown in Figure 3, the average activation energy *E* of the pyrolysis process of the waste compound was 237.05 kJ/mol. In addition, *E* varied greatly with α, particularly in the range of α from 0.1 to 0.4. It is worth noting that *E*, as a physical quantity representing the minimum energy required to start the pyrolysis reaction, is of great significance in the process of thermo-kinetic analysis. The drastic variation of *E* with α results in the occurrence of several reactions [55,56]. Therefore, it is emphasized once again that the single reaction model listed in Table 2 for characterizing simple reactions (or one-component reactions) makes it difficult to describe the pyrolysis process of complex rubber products clearly and accurately. However, as not all the additives participate in the reaction, and the content of the individual additives was relatively low, the effect of the additive reactions on the entire reaction was limited. The amount of additives is also insufficient to alter the characteristics for the type of reaction. As a result, the reaction temperature intervals of the different reaction types were not completely overlapped. Alternatively, the overlap of the reaction temperature range of the additives and the reaction temperature range of the rubber does not affect the chosen reaction type nor the reaction mechanism. Moreover, regarding the pyrolysis of the tread rubber, if there is a reaction mechanism model that summarizes the reaction types of all the additives in the tread rubber, it will be the most ideal result. After all, the reaction mechanism of the tread rubber obtained by thermogravimetric analysis of the sample is the main purpose of this study. There are three reaction stages in the literature to describe the pyrolysis of waste tires. Furthermore, the judgment made from the thermogravimetric curve only satisfies special conditions and does not have generality [57]. Conesa et al. also reached a similar conclusion that a fractional model is insufficient to explain the waste tire pyrolysis, but did not give a model to explain the first decomposition [39].

The average values of *E* for Reaction I and II were 155.26 and 315.40 kJ/mol, respectively, as shown in Figure 4. The arithmetic mean of *E* for Reaction I and II was calculated to be 235.33 kJ/mol, which is almost identical to the activation energy (237.05 kJ/mol) calculated based on the conversion curve of Figure 2a, as shown in Figure 3. Therefore, given the similarity of the values, a kinetic analysis based on Reaction I and II may be reasonable. Kim et al. studied the activation energy of sidewall rubber (147.03 kJ/mol) and tread rubber (128 kJ/mol) by the single scan rate method [58].

### 3.3. Reaction Mechanism Analysis

As mentioned previously in Section 2.3.1, a reaction model/mechanism to describe the pyrolysis process of the tread rubber cannot be found in Table 2. However, if the pyrolysis process of the tread rubber is divided into Reactions I and II (according to Section 2.3.1), and the kinetic analysis is carried out according to the methods of Section 2.3.2 and Section 2.3.3, then the reaction models/mechanisms in Table 2 can be used to characterize Reaction I and Reaction II.

It was predicted in Section 3.1 that Reaction I may conform to the power-law model listed in Table 2, and this can be proven through the experimental data curve of Figure 5a, which was plotted using the Malek method and standard kinetic equations given in Table 2. The standard kinetic equations conforming to the linear relationship in Figure 5 and the specific reaction model test results are listed in Table 3. According to the theory mentioned in Section 2.3.3, the slope of the curve of ln(g(α)) against lnβ at multiple heating rates should be equal to –1, and the corresponding linear correlation coefficient *R*^2^ should be equal to 1. The test results show that Reaction A can be described by the standard kinetic equation (Equation (35)) and conforms to the three-dimensional diffusion mechanism. From Table 3, except Reaction A, the test results of other reactions are not satisfactory. However, this does not mean that the reaction mechanism obtained by the Malek method is incorrect. On the contrary, the slope of the curve of ln(g(α)) versus lnβ is quite different from –1, which means that the coefficient of the reaction mechanism equation needs to be adjusted through a model correction factor. 

Figure 5b,d show that the standard kinetic equations (Equations (15)–(22)) have an ideal linear relationship with Reaction II and B. They can be described by the same reaction mechanism. It can be seen that the Malek method was very convenient and shows a simpler possible reaction mechanism, but it is not ideal for determining the specific reaction model. Therefore, the model checking method mentioned in Section 2.3.3 is important and a necessary procedure. The equations with this linear relationship and the corresponding test results are listed in Table 3. Similarly, it can be found that only the slope of the curve of ln(g(α)) versus lnβ (0.90) is not much different to –1. This is seen with Equation (20) listed in Table 2, which indicates that the reaction mechanism of Reaction II and B conform to random nucleation and subsequent growth. It also shows that, akin to Reaction I, the coefficients of the reaction mechanism equation need to be adjusted.

### 3.4. Model Reconstruction

In Section 2.3.3, it was mentioned that the pre-exponential factor *A*, which is significantly affected by G(α), can be obtained by the FWO method. At the same time, the analyses in Section 3.3 suggest that the 36 kinetic models listed in Table 2 cannot accurately determine Reaction I and Reaction II, so the reaction model needs to be modified [29,30,48]. According to Equation (1), it is known that *E* and *A* can be used to rearrange the reaction rate curve, dα/dT, in order to obtain the theoretical f(α) curve for the purpose of model correction. In this method, Equation (1) can be used to obtain the reaction model f(α), which characterizes the tread rubber of the silica-filled tire, and its validity is checked by comparing with experimental data. However, up to now, an accurate pre-exponential factor *A* is yet to be obtained. As such, it is necessary to select a relatively reliable G(α) from Table 3 to obtain a set of pre-exponential factors using Equation (6) and the activation energy. Finally, the corrected value of *A* is obtained with the modified model.

It was found in Section 3.3 that the reaction models of Reaction I and Reaction II are characterized by the power-law model and the J-A-M equation [51], respectively, as shown in Table 4. After correcting the model using the method above, the experimental data of Reactions I and II were compared to the modified power-law model, and the results are shown in Figure 6. The modified reaction models of Reaction I, II, and B, as well as the results of the method found by Section 2.3.3, are shown in Table 5.

Based on the most probable modified model functions G(α) for Reaction I and II (see Table 5 for details) and *E* obtained in Section 2.3.2, the FWO method was used to calculate *A* of the reactions, and the results are shown in Table 6. The average values of *A* for Reaction I and II are 7.7317 × 10^13^ S^−1^ and 1.7503 × 10^24^ S^−1^, respectively, while they are 2.3657 × 10^21^ S^−1^ and 8.1049 × 10^22^ S^−1^ for Reaction A and B, respectively. It was believed that when the obtained reaction model *G*(*α*) was suitable for characterizing solid pyrolysis, a linear relationship called the “compensation effect” existed between *E* and ln(A) [48,59], as shown in Equation (10). Figure 7 also shows that there is a strong linear relationship between ln(*A*) and *E* in all Reactions I, II, A, and B. The results once again demonstrate that the modified reaction models for characterizing Reaction I, II, and B are suitable.
(9)lnA=a+bE, 
where a and b are constants.

The average values of *E* and *A* for Reactions (I, II, A, B) are shown in Table 6. The study of the pyrolysis kinetic model and reaction mechanism of similar waste polymers can assist in the thermal management of pyrolysis engineering. In addition, the reasonable control of the process parameters can improve the efficiency of obtaining recycled raw materials. Therefore, values of the pyrolysis temperature range, activation energy, and pre-exponential factor of the tire tread rubber from existing literature are provided here and compared to the current work, as shown in Table 6.

### 3.5. Reaction Principle Analysis

Finally, through this research, it was found that the reaction model of Reaction I conforms to the power-law model, and its corresponding kinetic equation is $f\left(\alpha \right)=0.2473\alpha^{-3.0473}$, and the corresponding reaction mechanism conforms to the nucleation mechanism. Khawam and Flanagan proposed that the nucleation mechanism represents the formation of new product phases at certain reaction points (nucleation sites) in the reactant lattice [61]. The nucleation point indicates that the crystal has a fluctuating local energy due to defects such as impurities, surfaces, edges, dislocations, cracks, or point defects, where the activation energy of the reaction is minimized. Meanwhile, the tread rubber in the silica-filled tire contained uniform and non-pyrolytic fillers, as well as SiO_2_ and ZnO that have good heat transfer performance, resulting in quickly absorbed heat. In the case of insufficient heat, poor heat transfer performance, and irregularly directed heat transfer of the sample compound, the fillers exposed to the surface of the sample compound will first absorb enough heat and become the reaction point (nucleation point). Rubber chains in contact with fillers such as SiO_2,_ or those connected by chemical bonds will first absorb enough energy to break the chemical bonds, i.e., this portion of rubber macromolecular branched chains will undergo one-dimensional pyrolysis along the direction of branched chains. Senneca et al. also believed that the pyrolysis of waste tires can be divided into two main stages, of which the primary pyrolysis includes main chain scission and depolymerization [62]. Cherbański et al. observed that three characteristic steps can be distinguished from of the waste tire pyrolysis. These steps correspond to the evaporation and thermal decomposition of 1) oil, plasticizer, additives, and moisture in the first step, 2) natural rubber in the second step, and 3) styrene-butadiene rubber in the third step [60]. As a comparison, the analysis results from the inner liner using the same method show that Reaction A conforms to the three-dimensional diffusion mechanism. Combining the above conclusions and comparing the formulations for the two compounds, it is found that the system in which rubber and filler interact together is the determinant of the kinetic equation. Furthermore, when the main pyrolysis materials are natural rubber, by comparing Reaction II and Reaction B, it is found that the non-pyrolytic filler changes the parameters of the kinetic equation by changing the heat transfer mode. The schematic diagram of the pyrolysis of Reaction I is shown in Figure 8.

The reaction model of Reaction II can be explained by $f\left(\alpha \right)=0.4142\left(1-\alpha \right){\left[-\ln\left(1-\alpha \right)\right]}^{-1.4143}$, and the corresponding reaction mechanism of the equation is random nucleation and subsequent growth. The thermogravimetric experiment was carried out under a nitrogen atmosphere to avoid oxidation, and, as such, the pyrolysis reaction process was relatively clear. Intra-particle transport, which influences the global rate during pyrolysis, is caused by the increase in particle size [63], and an increase in particle size can subsequently increase the temperature gradient inside the particles [64]. Influenced by the industrial refining process of the tire tread rubber, the heat absorption and heat transfer efficiency of SiO_2_ and non-pyrolytic fillers are higher than those of the rubber material when they are mixed evenly in the compound. Therefore, the macromolecule chains wrapped around the filler will be gradually interrupted and decompose locally when the thermal energy is sufficient. However, the distribution of fillers with a high heat transfer efficiency is random, and the uniform distribution is affected by the rubber refining process. Thus, the influence of fillers on the direction of heat transfer tends to be irregular and random. When heat is transferred to the interior of the sample, the relative position between adjacent fillers will affect the heat transfer, therefore determining the reaction point (nucleation point) of the pyrolysis. In addition, as the surrounding heat increases, the macromolecular chains exposed to the surface gradually absorb enough energy to carry out the pyrolysis reaction due to edges, cracks, or point defects. It is due to these reasons that the position and time of the pyrolysis reaction are random. Ultimately, the compound on the surface of the sample undergoes a pyrolysis reaction with the gradual increase in abundance of the surrounding heat, and the macromolecular chain originally buried inside the sample is exposed to the surface to start the thermal cracking reaction. At the same time, because the macromolecular chain of the vulcanized rubber is a complex and irregular three-dimensional network structure, the tread rubber presents irregular heat transfer, therefore leading to Reaction II presenting random nucleation and subsequent growth. The schematic diagram of the Reaction II pyrolysis is shown in Figure 9.

## 4. Conclusions

When considering the pyrolysis of a material with complex components such as the tread rubber of a silica-filled tire, a single kinetic method often cannot obtain an accurate kinetic reaction mechanism and its corresponding factors. A reasonable and effective use of multi-kinetic methods, especially methods with model detection and model revision, can obtain pyrolysis kinetic equations and reaction mechanisms of such materials from different angles. It should be emphasized that this method does not have to assume the reaction model in advance to avoid unnecessary errors. Moreover, the correction and detection component of the kinetic function included in the method improves the confidence of the result. The kinetic results of this compound are listed in Table 6, and different reaction types can be controlled according to the activation energy and the temperature range of the reaction. The results show that the pyrolysis reaction process of this compound can be divided into Reaction I and Reaction II by the research methods proposed in this paper. In the case of insufficient thermal energy, the revised kinetic model, $f\left(\alpha \right)=0.2473\alpha^{-3.0473}$, and the nucleation mechanism can accurately characterize Reaction I. During this stage, non-pyrolytic fillers such as SiO_2_ and ZnO will take the lead in becoming nucleation sites for the pyrolysis reactions to occur. Moreover, thermal energy is transferred along the direction of the molecular chain, resulting in a one-dimensional fracture of the molecular chain. With the increase in thermal energy, the revised kinetic model, $f\left(\alpha \right)=0.4142\left(1-\alpha \right){\left[-\ln\left(1-\alpha \right)\right]}^{-1.4143},$ and the reaction mechanism of random nucleation and its subsequent growth can accurately characterize Reaction II. In comparison to the previously fixed nucleation sites, the appearance of new nucleation sites is random but mainly appears around randomly distributed fillers and exposed molecular chain edges. Future research will focus on the establishment of a pyrolysis reaction simulation system based on the three kinetic factors obtained and the kinetic reaction mechanism.

## Figures and Tables

**Figure 1 polymers-12-00810-f001:**
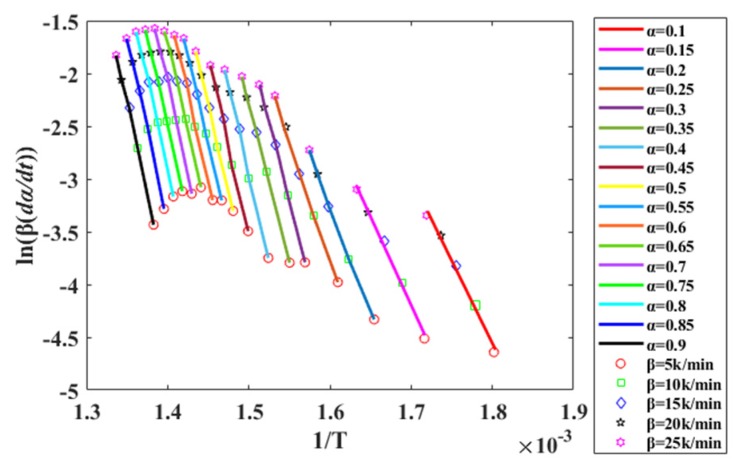
ln(βdα/dT) versus 1/T fit line for equal conversion rates at multiple heating rates.

**Figure 2 polymers-12-00810-f002:**
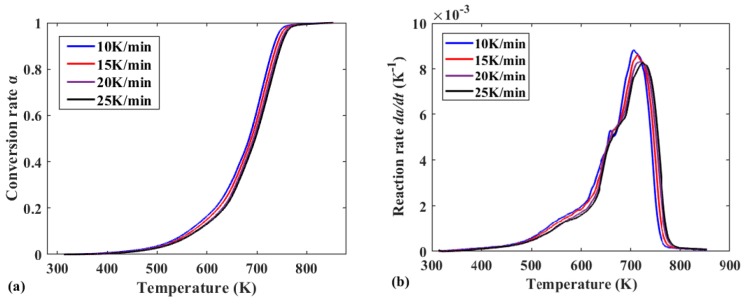
(**a**) Conversion curve of the entire reaction at different heating rates; (**b**) reaction rate curve of the entire reaction at different heating rates; (**c**) conversion curve of Reaction I at different heating rates; (**d**) reaction rate curve of Reaction I at different heating rates; (**e**) conversion curve of Reaction II at different heating rates; (**f**) reaction rate curve of Reaction II at different heating rates.

**Figure 3 polymers-12-00810-f003:**
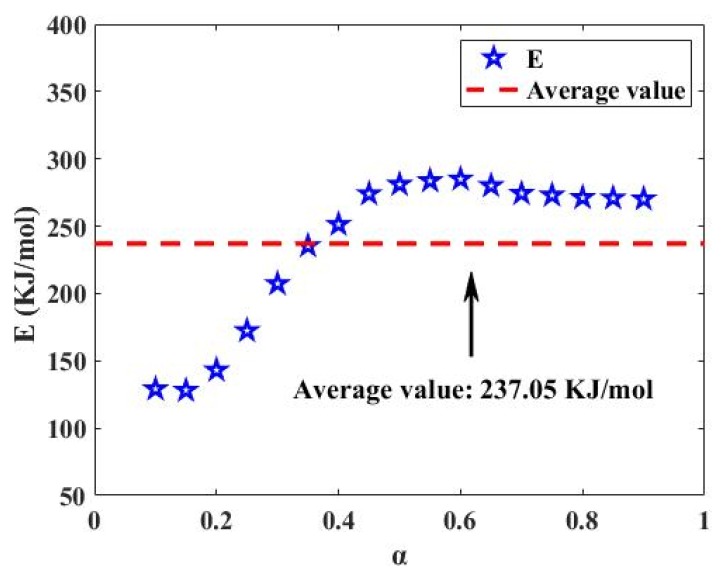
The *E–α* diagram of the whole reaction is based on Figure 2a.

**Figure 4 polymers-12-00810-f004:**
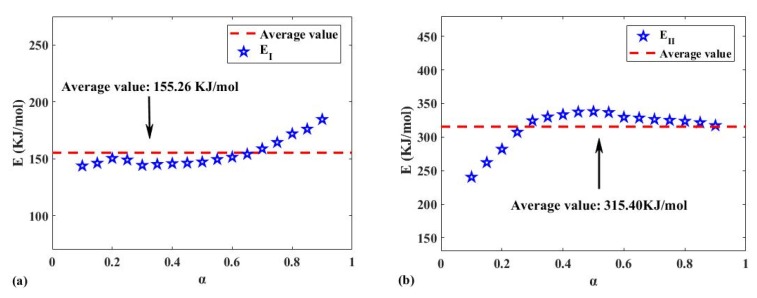
(**a**) The *E*–*α* diagram of Reaction I based on Figure 2c; (**b**) the *E*–*α* diagram of Reaction II based on Figure 2e.

**Figure 5 polymers-12-00810-f005:**
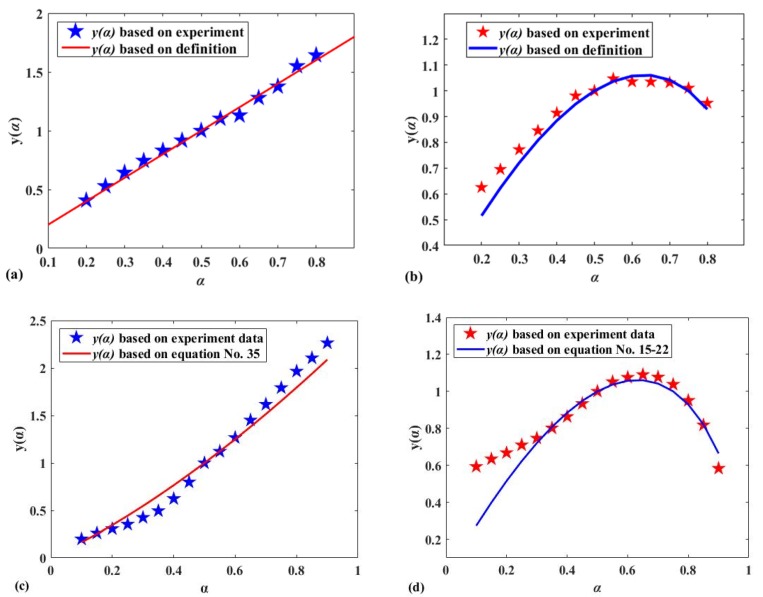
(**a**) Comparison of standard y(α) based on the equations listed in Table 3 with y(α) based on Reaction I; (**b**) comparison of standard y(α) based on the equations listed in Table 3 with y(α) based on Reaction II; (**c**) comparison of standard y(α) based on the equations listed in Table 3 with y(α) based on Reaction A; (**d**) comparison of standard y(α) based on the equations listed in Table 3 with y(α) based on Reaction B.

**Figure 6 polymers-12-00810-f006:**
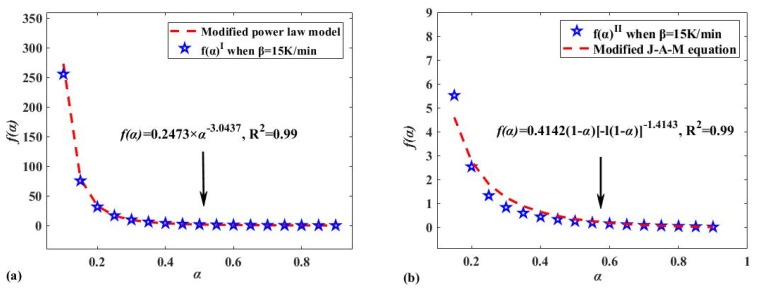
(**a**) The reconstructed experimental kinetic function f(α) of Reaction I compared to experimental data; (**b**) the reconstructed experimental kinetic function f(α) of Reaction II compared to experimental data.

**Figure 7 polymers-12-00810-f007:**
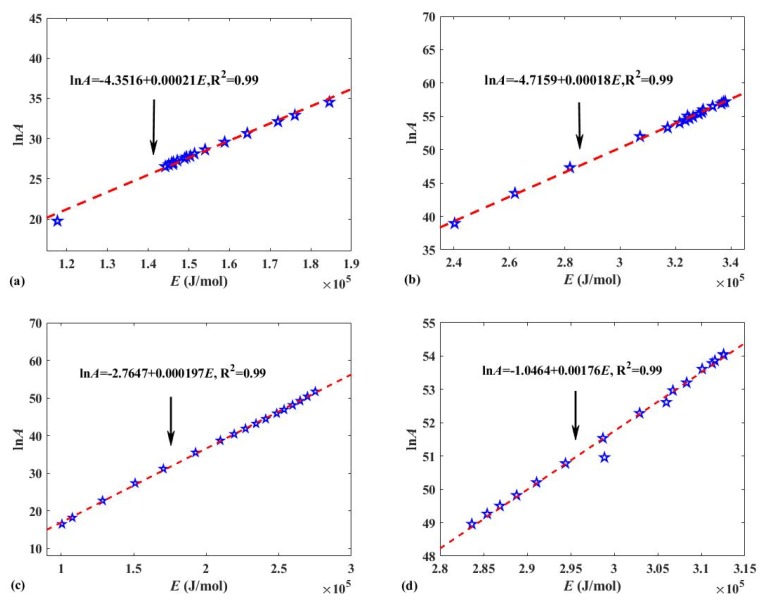
(**a**) Compensation effect between ln(*A*) and *E* of Reaction I; (**b**) compensation effect between ln(*A*) and *E* of Reaction II; (**c**) compensation effect between ln(*A*) and *E* of Reaction A; (**d**) compensation effect between ln(*A*) and *E* of Reaction B.

**Figure 8 polymers-12-00810-f008:**
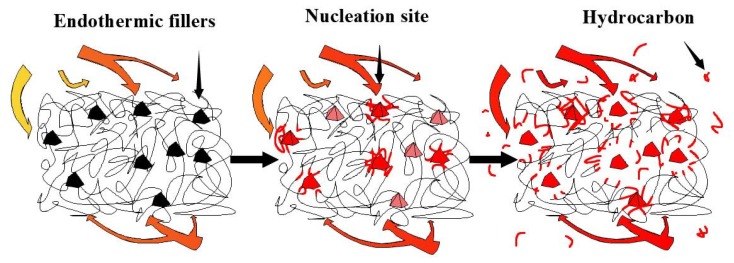
Schematic diagram of the pyrolysis of Reaction I.

**Figure 9 polymers-12-00810-f009:**
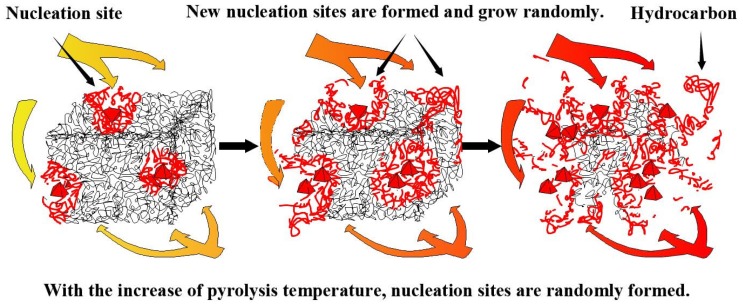
Schematic diagram of the pyrolysis of Reaction II.

**Table 1 polymers-12-00810-t001:** Formulation of the test samples.

**Tread rubber**	**Mixed ingredients (PHR)**						
	**SBR**	**NR**	**TSR20**	**Silica**	**V700**	**N234**	**Antilux**
	15	105	20	70	13	18	3
	**Si69**	**SAD**	**DPG**	**ZnO**	**S**	**CZ**	**4040**
	13	3	1	2	1.15	2	2
**Inner liner**	**Mixed ingredients (PHR)**						
	**BR9000**	**SBR**	**TSR20**	**N234**	**TDAE**	**RD**	
	15	15	70	45	3	1.5	
	**SAD**	**ZnO**	**Antilux**	**6PPD**	**CBS**	**S**	
	3	3.5	1.5	2	1.5	1.8	

Note: PHR - parts per hundreds of rubber, SBR - styrene butadiene rubber, NR - natural rubber, TSR - technical standard natural rubber, Silica - silicon dioxide, V700 - aromatic oils meeting EU standards, N234 - carbon black, Antilux - ceresin wax, Si69 - silane coupling agent, SAD - stearic acid, DPG - 1,3-diphenylguanidine, CZ - N-cyclohexyl-2-benzothiazolylsulfenamide, 4020 - Antioxidant 4020, BR9000 - butadiene rubber, TDAE - treated distillate aromatic extract, RD - Antioxidant RD, Antilux - ceresin wax, 6PPD - N-(1,3-dimethylbutyl)-N’-phenyl-p-phenylenediamine, CBS - N-cyclohexyl-2-benzothiazolylsulfenamide.

**Table 2 polymers-12-00810-t002:** The reaction model and mechanism used to describe solid-state pyrolysis [51].

No.	g(α)	f(α)	Rate-Determining Mechanism
1. Chemical process or mechanism non-invoking equations			
1	$1-{\left(1-\alpha \right)}^{2/3}$	$3/2{\left(1-\alpha \right)}^{1/3}$	Chemical reaction
2	$1-{\left(1-\alpha \right)}^{1/4}$	$4{\left(1-\alpha \right)}^{3/4}$	Chemical reaction
3	${\left(1-\alpha \right)}^{-1/2}-1$	$2{\left(1-\alpha \right)}^{3/2}$	Chemical reaction
4	${\left(1-\alpha \right)}^{-1}-1$	${\left(1-\alpha \right)}^{2}$	Chemical reaction
5	${\left(1-\alpha \right)}^{-2}-1$	$1/2{\left(1-\alpha \right)}^{3}$	Chemical reaction
6	${\left(1-\alpha \right)}^{-3}-1$	$1/3{\left(1-\alpha \right)}^{4}$	Chemical reaction
7	$1-{\left(1-\alpha \right)}^{2}$	1/2(1−α)	Chemical reaction
8	$1-{\left(1-\alpha \right)}^{3}$	$1/3{\left(1-\alpha \right)}^{2}$	Chemical reaction
9	$1-{\left(1-\alpha \right)}^{4}$	$1/4{\left(1-\alpha \right)}^{3}$	Chemical reaction
2. Acceleratory rate equations			
10	$\alpha^{3/2}$	$2/3\alpha^{-1/2}$	Nucleation
11	$\alpha^{1/2}$	$2\alpha^{1/2}$	Nucleation
12	$\alpha^{1/3}$	$3\alpha^{2/3}$	Nucleation
13	$\alpha^{1/4}$	$4\alpha^{3/4}$	Nucleation
14	lnα	α	Nucleation
3. Sigmoidal rate equations or random nucleation and subsequent growth			
15	−ln(1−α)	1−α	Assumed random nucleation and its subsequent growth
16	${\left[-\ln\left(1-\alpha \right)\right]}^{2/3}$	$3/2\left(1-\alpha \right){\left[-\ln\left(1-\alpha \right)\right]}^{1/3}$	Assumed random nucleation and its subsequent growth
17	${\left[-\ln\left(1-\alpha \right)\right]}^{1/2}$	$2\left(1-\alpha \right){\left[-\ln\left(1-\alpha \right)\right]}^{1/2}$	Assumed random nucleation and its subsequent growth
18	${\left[-\ln\left(1-\alpha \right)\right]}^{1/3}$	$3\left(1-\alpha \right){\left[-\ln\left(1-\alpha \right)\right]}^{2/3}$	Assumed random nucleation and its subsequent growth
19	${\left[-\ln\left(1-\alpha \right)\right]}^{1/4}$	$4\left(1-\alpha \right){\left[-\ln\left(1-\alpha \right)\right]}^{3/4}$	Assumed random nucleation and its subsequent growth
20	${\left[-\ln\left(1-\alpha \right)\right]}^{2}$	$1/2\left(1-\alpha \right){\left[-\ln\left(1-\alpha \right)\right]}^{-1}$	Assumed random nucleation and its subsequent growth
21	${\left[-\ln\left(1-\alpha \right)\right]}^{3}$	$1/3\left(1-\alpha \right){\left[-\ln\left(1-\alpha \right)\right]}^{-2}$	Assumed random nucleation and its subsequent growth
22	${\left[-\ln\left(1-\alpha \right)\right]}^{4}$	$1/4\left(1-\alpha \right){\left[-\ln\left(1-\alpha \right)\right]}^{-3}$	Assumed random nucleation and its subsequent growth
23	lnα/(1−α)	α/(1−α)	Branching nuclei
4. Deceleratory rate equations			
4.1. Phase boundary reaction			
24	α	${\left(1-\alpha \right)}^{0}$	Contracting disk
25	$1-{\left(1-\alpha \right)}^{1/2}$	$2{\left(1-\alpha \right)}^{1/2}$	Contracting cylinder (cylindrical symmetry)
26	$1-{\left(1-\alpha \right)}^{1/3}$	$3{\left(1-\alpha \right)}^{2/3}$	Contracting sphere (spherical symmetry)
4.2. Based on the diffusion mechanism			
27	$\alpha^{2}$	1/(2α)	One-dimensional diffusion
28	${\left[1-{\left(1-\alpha \right)}^{1/2}\right]}^{1/2}$	$4{\left\{\left(1-\alpha \right){\left[1-(1-\alpha )\right]}^{1/2}\right\}}^{1/2}$	Two-dimensional diffusion
29	α+(1−α)ln(1−α)	${\left[-\ln\left(1-\alpha \right)\right]}^{-1}$	Two-dimensional diffusion
30	${\left[-\ln{\left(1-\alpha \right)}^{1/3}\right]}^{2}$	$\left(3/2\right){\left(1-\alpha \right)}^{2/3}{\left[1-{\left(1-\alpha \right)}^{1/3}\right]}^{-1}$	Three-dimensional diffusion, spherical symmetry
31	$1-2/3\alpha -{\left(1-\alpha \right)}^{2/3}$	$\left(3/2\right){\left[{\left(1-\alpha \right)}^{-1/3}-1\right]}^{-1}$	Three-dimensional diffusion, cylindrical symmetry
32	${\left[{\left(1-\alpha \right)}^{-1/3}-1\right]}^{2}$	$\left(3/2\right){\left(1-\alpha \right)}^{4/3}{\left[{\left(1-\alpha \right)}^{-1/3}-1\right]}^{-1}$	Three-dimensional diffusion
33	${\left[{\left(1+\alpha \right)}^{1/3}-1\right]}^{2}$	$\left(3/2\right){\left(1+\alpha \right)}^{2/3}{\left[{\left(1+\alpha \right)}^{1/3}-1\right]}^{-1}$	Three-dimensional diffusion
34	$1+2/3\alpha -{\left(1+\alpha \right)}^{2/3}$	$\left(3/2\right){\left[{\left(1+\alpha \right)}^{-1/3}-1\right]}^{-1}$	Three-dimensional diffusion
35	${\left[{\left(1+\alpha \right)}^{-1/3}-1\right]}^{2}$	$\left(3/2\right){\left(1+\alpha \right)}^{4/3}{\left[{\left(1+\alpha \right)}^{-1/3}-1\right]}^{-1}$	Three-dimensional diffusion
36	${\left[1-{\left(1-\alpha \right)}^{1/3}\right]}^{1/2}$	$6{\left(1-\alpha \right)}^{2/3}{\left[1-{\left(1-\alpha \right)}^{1/3}\right]}^{1/2}$	Three-dimensional diffusion

**Table 3 polymers-12-00810-t003:** The calculation results by Malek method based on Figure 5 and the test results.

Reaction I	**No. of equation**		**10**		**11**	**12**		**13**		**27**
	Test results	Slope	−0.36		−0.12	−0.08		−0.06		−0.48
		R^2^	0.98		0.98	0.98		0.98		0.98
Reaction II	**No. of equation**		**15**	**16**	**17**	**18**	**19**	**20**	**21**	**22**
	Test results	Slope	−0.45	−0.30	−0.22	−0.14	−0.11	−0.90	−1.34	−1.79
		R^2^	0.99	0.99	0.99	0.99	0.99	0.99	0.99	0.99
Reaction A	**No. of equation**		**35**							
	Test results	Slope	−1.07							
		R^2^	0.99							
Reaction B	**No. of equation**		**15**	**16**	**17**	**18**	**19**	**20**	**21**	**22**
	Test results	Slope	−0.64	−0.43	−0.32	−0.21	−0.16	−1.28	−1.92	−2.56
		R^2^	0.99	0.99	0.99	0.99	0.99	0.99	0.99	0.99

**Table 4 polymers-12-00810-t004:** Power-law model and the J-A-M equation.

Function Name	Mechanism	Integral Form G(α)	Differential Form f(α)
Power law model	Acceleration α−t curve, Nucleation	$\alpha^{1/n}$	$n{\left(\alpha \right)}^{\left(n-1\right)/n}$
J-A-M equation	Assumed random nucleation and its subsequent growth	${\left[-\ln\left(1-\alpha \right)\right]}^{1/n}$	$n\left(1-\alpha \right){\left[-\ln\left(1-\alpha \right)\right]}^{1-1/n}$

**Table 5 polymers-12-00810-t005:** Modified power-law model and the J-A-M equation.

Reaction	Model	Integral Form G(α)	Differential Form f(α)	n	R^2^	Result Test	
						Slope	R^2^
I	Power-law model	$\alpha^{4.0473}$	$0.2473{\left(\alpha \right)}^{-3.0437}$	0.2473	0.99	−0.98	0.98
II	J-A-M equation	${\left[-\ln\left(1-\alpha \right)\right]}^{2.4143}$	$0.4142\left(1-\alpha \right){\left[-\ln\left(1-\alpha \right)\right]}^{-1.4143}$	0.4142	0.99	−1.08	0.99
B	J-A-M equation	${\left[-\ln\left(1-\alpha \right)\right]}^{1.8501}$	$0.5405\left(1-\alpha \right){\left[-\ln\left(1-\alpha \right)\right]}^{-0.8518}$	0.5405	0.99	−1.00	0.99

**Table 6 polymers-12-00810-t006:** Comparison of the conclusions obtained from this study with other published results.

Research Object	Degradation Temperature Range (K)	Activation Energy (kJ/mol)	Pre-Exponential Factor (1/min)	Source
Reaction I	500–645	155.26	1.29 × 10^12^	This study
Reaction II	645–750	315.40	2.92 × 10^22^	
Tread rubber	500–750	237.05	2.92 × 10^22^	
Reaction A	515–650	219.89	3.94 × 10^19^	
Reaction B	650–720	300.58	1.35 × 10^21^	
Inner liner	515–720	280.72	1.35 × 10^21^	
Tread rubber of unknown tire	300–773	33–283	7.56 × 10^2^–1.39 × 10^19^	[2,60]

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
