# Peer review of "Study on the Pyrolysis Kinetics and Mechanisms of the Tread Compounds of Silica-Filled Discarded Car Tires"

_polymers, 2020, doi:10.3390/polym12040810_

Round 1

Reviewer 1 Report

Submitted paper has a very good quality. I suggest its publishing as it is without any changes.

Author Response

Dear Professor

First of all, thank you very much for your approval of the manuscript content. Your comment inspires our researchers and makes us passionate at work. Wish you a happy life.

Reviewer 2 Report

The authors reported a research on kinetic study of pyrolysis of waste plastics and tires which is interesting. Their aim is to develop a kinetic model and estimate the parameters that could explain the mechanism. The reaction is complex for a kinetic analysis but authors considered all limitations and made a comparison with previous reports.

This work is a comprehensive kinetic analysis and a useful contribution that meets the standards of journal of Polymers and should be published in present form.

Author Response

(The authors gave the same response as above.)

Reviewer 3 Report

The manuscript reports a kinetics analysis on TG data from a composite material. The limited number of samples and the very standard analysis make this work of low impact.

Conclusions on structure-properties correlations are not supported by the experiments as a larger number of samples are required.

Isoconversional methods are standard and well known, the description can be shorted a lot (the lenght of this MS is not justified).

Form the Activation energy vs alpha plots it is clear that Ea is not constant and therefore the analysis and applicability of some isoconversional methods should be reconsidered.

Reviewer 4 Report

The paper by Wang et al. proposes a multi-kinetic model aiming to shed light on the pyrolysis mechanism of very complex material, silica loaded tire tread. Findings might be useful for the waste management of such environmentally dangerous materials. Overall, the presented two-stage pyrolysis method looks interesting since it does not require model assumptions. However, the major and minor comments below must be addressed before acceptance.

Major Comments:

  • The major challenge of this paper is trying to explain a very complex pyrolysis mechanism of a very complex material by investigating only one tire tread sample. In such composite systems, there are many chemical parameters such as crosslinking density, filler amount, filler-polymer interaction, which might change the pyrolysis dynamics. Hypothesis in Figure 8 and Figure 9 is very important. However, it must be supported preferentially by further experiments (such as reference system without vulcanized or having different filler amounts) or introducing examples from literature to support the idea.
  • The design/content/use of Table 2 is very crucial for understanding the work. However, it is not very clear to follow the data presented in the table, especially when it is called in the text. Maybe using the “No” part (numbers) in the table more efficiently, decreasing the table content and using some references in the table might be useful.
  • In general, the author must explain what experimental or chemical factors can vary the precise kinetic equations (f(α)) of both Reactions (Reaction I and Reaction II). Again, it would be interesting to see data from different samples. Furthermore, it would be interesting to know if a single stage (kinetic equation) can ever explain the pyrolysis of the almost similar tire tread samples having small chemical variations (such as filler size, the density of crosslinking, etc.).
  • In the Introduction, the author used much jargon such as “pre-exponential factor,” “optimization algorithms,” “pre-supposition model.” It is not practical to use such terminologies unless the article is submitted to specific journals like J. Anal. Appl. Pyrolysis. For a journal like “Polymers,” the Introduction needs to be more general, and it should prepare the reader for the upcoming details in the manuscript. The author needs to explain the terms better in the Introduction without making the text longer. 

 Minor comments/questions/suggestions:

  • The author must prepare and independent “Graphical Abstract,” not the same illustration used in Figure9.
  • What are the following abbreviations GA, SCE, FWO, and the KAS?
  • What does the “corresponding physical model” in line 100 in p3 and “further screening” in line 214 in p8 mean?.
  • Sentence between lines 108-111 requires a reference.
  • It might be helpful to mention why silica used as filler but not carbon black?
  • Abbreviations used in Table 1 need to be explained in the Methods.
  • What does the “material particle size” in line 146 stand for? Filler size?
  • Generating (i) Fig. 2c, and Fig. 2e from Fig. 2a and (ii) Fig 2b,d,f from Fig 2a,c,e must be explained in the Methods, explicitly.
  • What does the “as a mature kinetic research method” mean in line 221 in p8.
  • What does “correct activation energy” means in line 279?
  • Connected to the Reviewer’s major comment 2, strong argumentations in the text below require either further experiments or better literature reviewing. What does the author mean by “the amount of additives is also insufficient”? Is there any limit between sufficient and insufficient with the additive amount? Does also the additive type (chemistry) taken into account in such a limit? “However, since not all the additives participate in the reaction, and the content of the individual additives was relatively low, the effect of the additive reactions on the entire reaction was limited. The amount of additives is also insufficient to alter the characteristics for the type of reaction.”
  • What are the σ in some Figures as Figure 3 and Figure 4. If (probably) they are standard deviation, the way to calculate them must be added to the Methods.
  • Why was the symbol “~” used in line 324? Also, what does “akin” means in line 349 in p13.
  • Please fix the typo in Figure 6b.
  • The author must specify the polymer type “tires” in Table 8. The reviewer is also not very sure if naming the very first column as “Polymer type” is correct or not?

Round 2

Reviewer 4 Report

In the opinion of the reviewer, the authors did a serious revision. Especially, supporting the argumentation by adding additional material and literature is important. 

A serious spell check before publication is recommended.